# Calibrating CNNs for Lifelong Learning

**Pravendra Singh**[*1], **Vinay Kumar Verma**[*2], **Pratik Mazumder**[1],
**Lawrence Carin**[2], **Piyush Rai**[1]
[1]CSE Department, IIT Kanpur, India    [2]Duke University, USA
psingh@cse.iitk.ac.in, vinaykumar.verma@duke.edu,
pratikm@cse.iitk.ac.in, lcarin@duke.edu, piyush@cse.iitk.ac.in

## Abstract

We present an approach for lifelong/continual learning of convolutional neural networks (CNN) that does not suffer from the problem of catastrophic forgetting when moving from one task to the other. We show that the activation maps generated by the CNN trained on the old task can be calibrated using very few calibration parameters, to become relevant to the new task. Based on this, we calibrate the activation maps produced by each network layer using spatial and channel-wise calibration modules and train only these calibration parameters for each new task in order to perform lifelong learning. Our calibration modules introduce significantly less computation and parameters as compared to the approaches that dynamically expand the network. Our approach is immune to catastrophic forgetting since we store the task-adaptive calibration parameters, which contain all the task-specific knowledge and is exclusive to each task. Further, our approach does not require storing data samples from the old tasks, which is done by many replay based methods. We perform extensive experiments on multiple benchmark datasets (SVHN, CIFAR, ImageNet, and MS-Celeb), all of which show substantial improvements over state-of-the-art methods (e.g., a 29% absolute increase in accuracy on CIFAR-100 with 10 classes at a time). On large-scale datasets, our approach yields 23.8% and 9.7% absolute increase in accuracy on ImageNet-100 and MS-Celeb-10K datasets, respectively, by employing very few (0.51% and 0.35% of model parameters) task-adaptive calibration parameters.

## 1 Introduction

Humans are adept at continual/lifelong learning of multiple tasks in a sequence, while not forgetting the knowledge acquired from earlier tasks when subjected to new learning tasks. Unfortunately, deep neural networks do not have such an inherent property. It has been well-recognized that deep neural networks suffer from the problem of catastrophic forgetting [1], e.g., in a categorization task, the network performance on the previously trained categories tends to fall drastically as if the network has "forgotten" those categories of data.

Continual learning [2, 3] involves training the network in such a way that it can continually learn from tasks that arrive sequentially and add to its existing knowledge base instead of replacing knowledge about the older tasks. A trivial solution to this is to simply store all the data from all the tasks as and when they arrive so that whenever a new task arrives, we can train the network on all the previous task data and the current task data. However, deep learning is used in many diverse applications, where storing such a large amount of data is not feasible. Therefore, an incrementally trained model must be able to learn from new tasks that arrive sequentially by only training on that task and still retain the knowledge gained from the previous task without having to re-train on all the previously

---

[*]Equal contribution.

seen data. Finally, we should obtain a model that performs well for all the sequentially added tasks. Solving this problem will make deep learning models much more human-like.

In addition to preserving the knowledge of the old tasks, lifelong learning models must also leverage this knowledge to help in learning new tasks. This is referred to as forward transfer. When training on a new task, if the network drastically forgets knowledge from the older tasks, then it is said to be a plastic network. On the other hand, if a network gives so much importance to the older tasks that it is unable to properly learn the new task, then it is said to be a stable network. Too much stability or plasticity will be harmful to this problem. Therefore, for lifelong learning (which we will occasionally interchangeably refer to as continual/incremental learning in the rest of the exposition) to be successful, a stability-plasticity balance must be maintained.

The following objectives are important for any lifelong learning algorithm: 1) the network should exhibit zero or near-zero catastrophic forgetting; 2) in the attempt to achieve this, the method performance should not fall significantly on the continual learning task; 3) the number of parameters and computations in the network should not increase significantly; and 4) the network should be able to perform forward knowledge transfer from old tasks to the new ones. Even when one achieves zero catastrophic forgetting, this does not trivially lead to good performance. For example, some existing methods such as Piggyback [4] and others [5, 6] do not suffer from catastrophic forgetting but suffer from performance degeneration and hence fail as per the second objective. On the other hand, some methods [7, 5, 8] increase the network size significantly and hence fail the third objective. To the best of our knowledge, none of the existing methods satisfy all of the above objectives.

Transfer learning is a technique for transferring knowledge from one dataset/domain/network to another [9]. It enables utilizing the knowledge gained from training the network on the source dataset to help in training the network better on the target dataset and also help converge faster. The intuition behind transfer learning is that the initial layers of the network typically learn to extract basic features common to both source and target datasets, such as edges and corners of an object (in image-based data). However, based on this idea, if we directly train the network on a new task, it will override the older parameter weights that had been learned for the older task, resulting in catastrophic forgetting. In order to prevent this, we can freeze the entire network that has been trained on the older task and only train extra network layers for each new task. This will eliminate catastrophic forgetting while promoting knowledge transfer from the initial task to the later task using the frozen network. However, adding extra network layers for each new task is not a scalable solution when the number of tasks is large. A more desirable approach would be to learn a small number of (re)calibration parameters to modify the intermediate activations produced by the frozen network to make them relevant to the new task. This will have no catastrophic forgetting and also only cause a very insignificant increase in network size per task.

We propose a novel method to address the lifelong learning problem in convolutional neural networks (CNNs) based on the above idea, aimed at accomplishing all of the aforementioned objectives. We focus on maximum re-use of features generated by the CNN trained on the first task to obtain features for images in the subsequent tasks. We achieve this by performing spatial and channel-wise calibration of the intermediate activation-maps according to the task that is currently being learned. Specifically, our method involves training a CNN-based base module for the first task and training a set of calibration parameters for every intermediate activation map generated by the base module for all subsequent tasks. Except for the first task, task-adaptive calibration parameters are the only trainable parameters. Moreover, the calibration parameters of the previous task also serve as good initial weights for learning the calibration parameters of the new task. Since our approach re-uses the CNN trained on the old task for the new tasks, forward knowledge transfer is achieved. During testing, when the task is changed, we simply change the calibration parameters. This ensures no catastrophic forgetting for the previous tasks and an insignificant increase of parameters and computation per task compared to other dynamic network-based continual learning methods [5, 10, 11, 12, 13, 14]. Our proposed method is described in detail in Sec 2.2. We perform extensive experiments on several benchmark datasets and perform various ablation experiments to validate our approach.

To summarize, our major contributions are as follows:

- We propose a novel method of lifelong learning for convolutional neural networks, which involves (re)calibrating the activation maps generated by the network trained on older tasks to produce features relevant to the newer tasks.

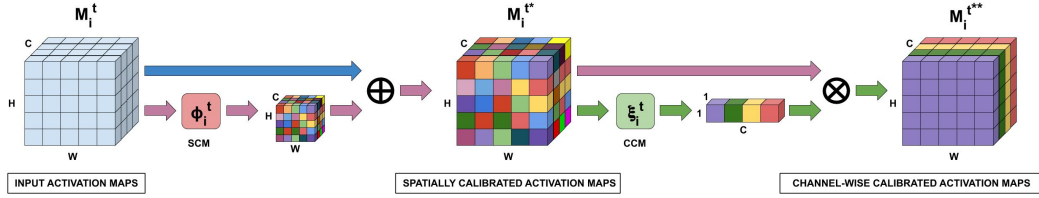

Figure 1: Calibration module (CM) containing the spatial calibration module (SCM) and channel-wise calibration module (CCM) that are applied sequentially to the activation maps. Here $\oplus$ and $\otimes$ represent element-wise addition and channel-wise multiplication operation respectively.

- We empirically show that our method introduces a very small number of parameters compared to other dynamic network-based lifelong learning methods.

- We experimentally show that our method performs significantly better than existing, state-of-the-art lifelong learning methods.

## 2 Proposed Method

### 2.1 Problem Setting

We consider the task incremental classification setting [7], where new tasks with new sets of classes are sequentially provided to the network. Let the total number of tasks be $K$, each having $U$ new classes. The objective is to train a network in this setting such that the final network performs well on the new tasks as well as on the old tasks without any performance loss.

### 2.2 Method Overview

As mentioned earlier, we propose a lifelong learning approach for convolutional neural networks called Calibrating CNNs for Lifelong Learning (CCLL). It is designed to effectively re-use the features learned by the network, trained on the initial task, and efficiently (re)calibrate them, using a very small number of calibration parameters, in order to make them relevant to the new tasks. Our method requires task-labels during test time in order to identify which task-adaptive calibration parameters to use for re-calibrating the convolutional layer outputs.

Our network consists of three types of modules: base module $E$, task-adaptive calibration modules $CM_i^t$, and task-specific classification modules $C^t$. The base module is a convolutional neural network with $N$ layers ($L_1$ to $L_N$) and parameters $\theta_E$. Each layer $i \in [1, N]$ produces an output activation map $M_i^t$, where $t$ refers to the task $t$. We add a calibration module (CM) after each layer of the base module. The calibration module $CM_i^t$ is added after the $i^{th}$ layer of the base module for task $t$. Each calibration module consists of a spatial calibration module (SCM) followed by a channel-wise calibration module (CCM), as shown in Fig. 1. The spatial calibration module learns weights to calibrate each point in the activation maps while the channel-wise calibration module learns weights to calibrate each channel of the activation maps. The output of the $i^{th}$ layer of the base module is fed to the $i^{th}$ calibration module, which feeds its output to the $(i + 1)^{th}$ layer of the base module.

Assume $M_i^t$ to be of size $H \times W \times C$, where $H$, $W$, and $C$ denotes height, width, and number of channels, respectively. Let $\Phi_i^t$ be the SCM operator added after the $i^{th}$ layer of the base module for task $t$. The spatial calibration module uses group convolution with $3 \times 3$ kernel size, the number of groups equal to $\frac{C}{\alpha}$ with each group having $\alpha$ channels. The output of $\Phi_i^t$ will also be of size $H \times W \times C$, representing the spatial calibration weights. Therefore, the SCM operator can be described as the function $\Phi_i^t : \mathbb{R}^{H \times W \times C} \longrightarrow \mathbb{R}^{H \times W \times C}$

$$\Phi_i^t(M_i^t) = \text{GCONV}_\alpha(M_i^t) \tag{1}$$

where $\text{GCONV}_\alpha$ represents group convolution with the number of groups equal to $\frac{C}{\alpha}$.

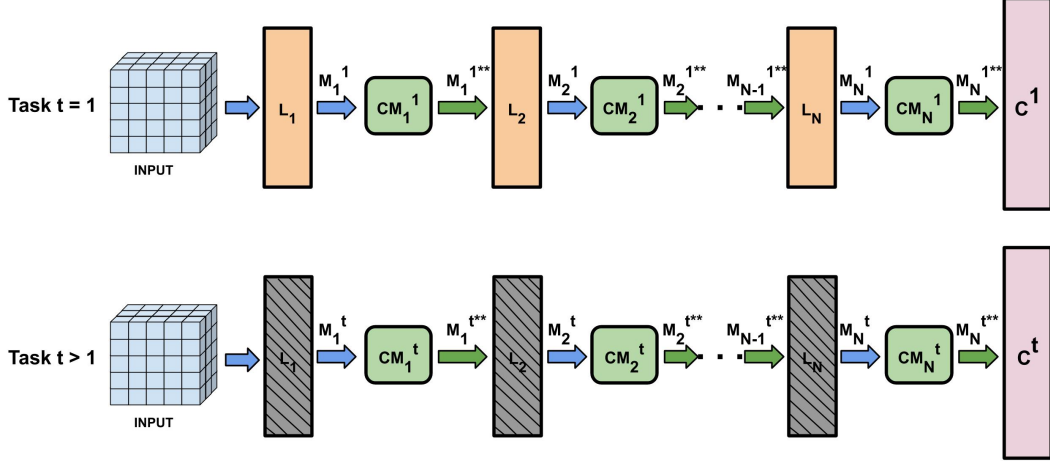

Figure 2: Our proposed architecture for lifelong learning. The top architecture is for the first task $t = 1$ and bottom architecture is for all the subsequent tasks $t > 1$. $L_1 - L_N$ represent the layers of the base module. The calibration module $CM_i^t$ calibrates the $i^{th}$ layer output $M_i^t$ to produce $M_i^{t**}$ that is given as an input to the $i + 1^{th}$ layer. After the first task, for all the subsequent tasks, the base module is frozen and its layers are not trainable and are marked in gray color with hatched pattern.

The calibration weights will get added element-wise to $M_i^t$ to give the spatially calibrated activation maps $M_i^{t*}$. The spatially calibrated activation maps $M_i^{t*}$ are given as input to the channel-wise calibration module. $M_i^{t*}$ is obtained as

$$M_i^{t*} = \Phi_i^t(M_i^t) \oplus M_i^t \tag{2}$$

where $\oplus$ represents the element-wise addition operation.

Let $\xi_i^t$ be the CCM operator added after the SCM operator for the $i^{th}$ layer of the base module for task $t$ as shown in Fig. 1. The channel-wise calibration module first performs global average pooling (GAP) on $M_i^{t*}$. This produces an output of size $1 \times 1 \times C$. The channel-wise calibration module performs group convolution with kernel size $1 \times 1$, the number of groups equal to $\frac{C}{\beta}$ with each group having $\beta$ channels, on the output of the global average pooling operation. This is followed by a sigmoid activation function that again produces an output size of $1 \times 1 \times C$ which represents the channel-wise calibration weights. Therefore, the CCM operator can be described as the function $\xi_i^t : \mathbb{R}^{H \times W \times C} \longrightarrow [0,1]^{1 \times 1 \times C}$

$$\xi_i^t(M_i^{t*}) = \sigma(BN(\texttt{GCONV}_\beta(GAP(M_i^{t*})))) \tag{3}$$

where $\texttt{GCONV}_\beta$ represents group convolution with the number of groups equal to $\frac{C}{\beta}$, BN represents batch normalization, and $\sigma$ represents the sigmoid activation function.

Each of the calibration weights gets multiplied to the corresponding channel of $M_i^{t*}$ to produce the final calibrated activation maps $M_i^{t**}$ for the $i^{th}$ layer. $M_i^{t**}$ can be obtained as

$$M_i^{t**} = \xi_i^t(M_i^{t*}) \otimes M_i^{t*} \tag{4}$$

where $\otimes$ represents the channel-wise multiplication operation.

Therefore, we can describe the overall calibration process as a combination of Eqs. 2 and 4 (Fig. 1)

$$M_i^{t**} = CM_i^t(M_i^t) = \xi_i^t(\Phi_i^t(M_i^t) \oplus M_i^t) \otimes (\Phi_i^t(M_i^t) \oplus M_i^t) \tag{5}$$

where $CM_i^t$ is the task $t$ calibration module added after the $i^{th}$ layer of the base module.

For the first task, we train the base module, the calibration modules, and the classification module. For the subsequent tasks $t > 1$, we keep the base module weights $\theta_E$ as frozen and only train the task-adaptive calibration modules $CM_i^t$ for all $i \in [1, N]$, and the task-specific classification module $C^t$. In this way, we adapt features relevant to the new task from the base module using the calibration

modules. We train the network only on the classification loss using cross-entropy loss, and we do not use the distillation loss or task exemplar replay/rehearsal. The full architecture is shown in Fig. 2.

Through an ablation experiment, we show that if we do not train the base module at all and only train the calibration parameters for each task, the network performance is hurt drastically. Therefore, in our method, transfer of knowledge occurs from the first task to the later tasks using the base module (that is only trained on the first task). Through another ablation experiment, we show that if we use the calibration parameter weights of the previous task as the initial weights for training the calibration parameters for the next task, we get better results than training the task adaptive calibration parameters from scratch for each task. This also shows that forward transfer of knowledge is happening from previous tasks to the next task.

The task-adaptive calibration parameters are stored. During testing, depending on the task-label, the corresponding task-adaptive calibration parameters are used, and classification is performed. Since our calibration module is light-weight, the number of extra parameters introduced per task and the increase in the total number of computations are not significant. Therefore, our proposed method is a very efficient lifelong learning method with no catastrophic forgetting.

## 3 Related Work

Continual learning methods can be broadly divided into three types: regularization-based, memory-based, and dynamic-network-expansion-based methods.

**Regularization based Methods:** These methods use regularization techniques to ensure that the network outputs do not change drastically while training on new tasks, so that the old task knowledge is preserved to some extent. The method described in [15] uses knowledge distillation to achieve the above goal. In [16, 17], the authors make use of distillation loss in addition to modified classification techniques suited to continual learning. The learning rate is decreased in [18] for the parameters that are important to the older tasks. In a similar spirit, [19] employs intelligent synapses that use task-relevant knowledge to store new task information while minimizing the loss of old task knowledge. Our method does not need to use regularization loss since it does not suffer from catastrophic forgetting.

**Memory based Approaches:** These methods store old task data or data representatives that are used during training for new tasks, so that catastrophic forgetting is reduced for older tasks (prototype rehearsal/replay). In [16], the authors use exemplar-based prototype rehearsal along with distillation to tackle forgetting. A similar approach is applied to a cross-dataset setting in [20]. The works in [21, 22, 23] focus on brain-inspired short and long-term memory with sleep phases. A custom architecture is used in [21, 23] to produce pseudo samples for the older tasks to be used for prototype rehearsal. Pseudo sample-based rehearsal is also used in [24], by making use of a generator and discriminator for generating such samples. In [25], the authors store the old task representative data and the class-specific statistics of those tasks for an improved prototype rehearsal. Optimization-based meta-learning with experience replay is used in [26]. Our method does not store any data from the previous tasks and, therefore, does not need any dedicated memory for storing task exemplars. We only store very few task-adaptive calibration parameters for new tasks.

**Dynamic Network Methods:** These methods dynamically modify the network to deal with training on new tasks, usually by network expansion. The method described in [5] creates a new neural network for each new task with lateral connections to the old task networks for forward knowledge transfer. A hierarchical expansion of the network is carried out in [10] to perform classification for both coarse super-classes, which have similar classes clustered together and full classification within the super-class. In [11], the authors use a tournament selection genetic algorithm to choose a subset of pathways through the network for the tasks, and they re-use relevant pathways for new tasks. The work in [12] selectively re-trains and dynamically expands the network for new tasks with only relevant neurons and also splits and duplicates neurons. In [27], the authors perform a weighted update on the network according to the episodic memory gradient. Reinforcement learning is used in [13] to decide the number of neurons to add for each new task. The method described in [14] predicts how much network weights to re-use and how much extra parameters to add for each new task. A random path selection methodology is proposed in [7] for faster convergence. It also uses a distillation procedure with exemplar-based rehearsal to provide a significant boost to the incremental learning performance. Network expansion for new tasks is kept limited in [8], and

network parameters are demarcated as shared and task-specific. The weights of the task-specific model are generated based on the task identity in [6].

The dynamic network methods have been shown to be among the most successful ones for lifelong learning, albeit the number of parameters in such methods can quickly become very large. Our feature map re-calibration based approach is motivated by dynamic methods but requires significantly less storage and computation, as we show through our experiments.

# 4 Experiments

## 4.1 Datasets

We perform experiments on the SVHN [28], CIFAR [29], ImageNet [30], and MS-Celeb-10K [31] datasets. In the case of SVHN, which has 10 classes, we group 2 consecutive classes to get 5 tasks, and we incrementally train on the five tasks. We perform experiments on CIFAR-100 with 10 tasks where each task contains 10 classes. For split CIFAR-10/100 experiments, we use all the classes of CIFAR-10 for the first task and randomly choose 5 tasks of 10 classes each from CIFAR-100. So we have 6 tasks for this setting. In the case of ImageNet-100, we use the subset proposed by [16] containing 100 classes, and we group them into 10 tasks of 10 classes each. In the case of MS-Celeb-10K, we use the subset of MS-Celeb consisting of 10000 classes [32], and we group them into 10 tasks of 1000 classes each.

## 4.2 Implementation Details

For SVHN and CIFAR-100 experiments, we use the ResNet-18 architecture. For split CIFAR-10/100 experiments, we use the ResNet-18 and ResNet-32 architectures. In the above experiments, we train the network for 150 epochs for each task with the initial learning rate equal to 0.01, and we multiply the learning rate by 0.1 at the 50,100 and 125 epochs. We also perform experiments with the LeNet architecture [33] on CIFAR-100. We train the network for 100 epochs for each task with the initial learning rate equal to 0.01, and we multiply the learning rate with 0.5 at the 20,40,60 and 80 epochs.

For ImageNet-100 experiments, we use the ResNet-18 architecture. We train the network for 150 epochs for each task with the initial learning rate equal to 0.01, and we multiply the learning rate by 0.1 at the 50,100 and 125 epochs. For MS-Celeb-10K experiments, we use the ResNet-18 architecture. We train the network for 70 epochs for each task with the initial learning rate equal to 0.01, and we multiply the learning rate by 0.1 at the 20,40 and 60 epochs. We use the SGD optimizer in all our experiments. In all cases, we run experiments for 5 random task orders and report the average accuracy.

We refer to our model as CCLL<$\alpha, \beta$>, where $\alpha$ and $\beta$ are the number of channels per group in the group convolution used in the spatial calibration module and channel-wise calibration module, respectively. We found that using $\beta > 1$ does not significantly improve the performance of our model. Therefore, in all our experiments, we set $\beta = 1$.

## 4.3 Experiments on Small-Scale Datasets

**SVHN:** Table 1 reports the experimental results on the SVHN dataset. From the results, we can see that our method CCLL<1,1> outperforms existing state-of-the-art methods significantly. CCLL achieves an absolute increase of about 9.3% over RPS [7].

**CIFAR-100:** For CIFAR-100 incremental learning tasks using 10 classes at a time, we compare our CCLL method with multiple state-of-the-art approaches: Learning without Forgetting (LwF) [15], Synaptic Intelligence (SI) [19], Elastic Weight Consolidation (EWC) [18], Incremental Classifier and Representation Learning (iCARL) [16] and Random Path Selection (RPS) [7]. Figure 3 shows the comparison of our method with the above-mentioned methods. We use ResNet-18 architecture in these experiments as used in [7]. From the graphs, we can see that our CCLL method performs better than all existing methods. Our CCLL<1,1> model outperforms RPS [7] by an absolute margin of 26% for the 10 classes per task. Our CCLL<4,1> model outperforms RPS [7] by an absolute margin of 29% for the 10 classes per task setting. As seen in Fig. 3, our method consistently performs better than all other methods as more tasks are seen. Therefore, our approach provides enough capacity for the network to learn new tasks.

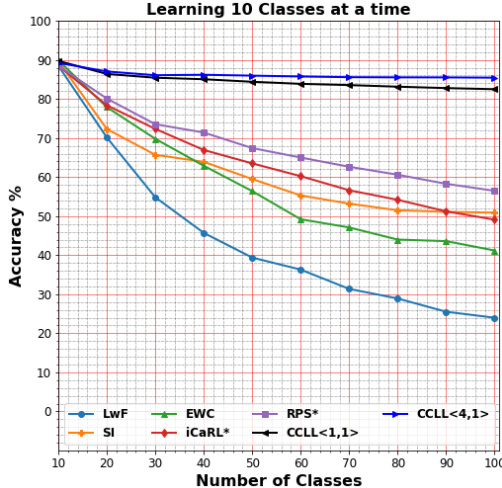

Figure 3: Experimental results on CIFAR-100 dataset with tasks containing 10 classes. '∗' denote memory based approaches.

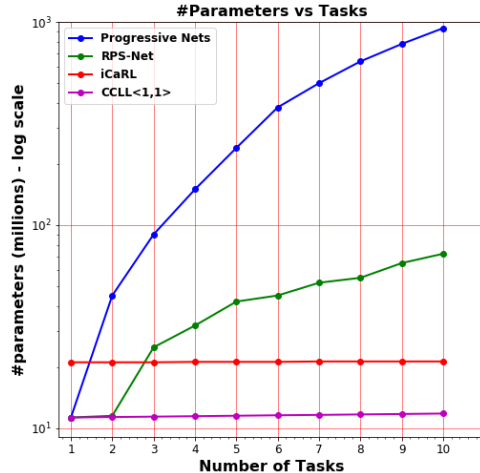

Figure 4: Increase in model parameters with number of tasks.

Table 1: Experimental results on SVHN dataset with ResNet-18 architecture. We report the average accuracy of 5 tasks ($A_5$). '∗' denote memory based approaches.

| Methods | SVHN ($A_5$) |
|---|---|
| GEM∗ [34] | 75.61% |
| RPS∗ [7] | 88.91% |
| CCLL<1,1> (Ours) | **98.20**% |

Table 2: Experimental results on CIFAR-100 with LeNet architecture.

| Methods | Capacity | Accuracy |
|---|---|---|
| STL [8] | 1000% | 63.75% |
| L2T [8] | 100% | 48.73% |
| EWC [18] | 100% | 53.72% |
| P&C [35] | 100% | 53.54% |
| PGN [5] | 171% | 54.90% |
| DEN [12] | 181% | 57.38% |
| RCL [13] | 181% | 55.26% |
| APD [8] | 135% | 60.74% |
| CCLL<1,1> (Ours) | 100.7% | **63.71**% |
| CCLL_BN<1,1> (Ours) | 101.6% | **71.52**% |

In Table 1, we compare CCLL with GEM [34] that uses task labels during testing as in our method. In Fig. 3, we compare CCLL with LwF [15], EWC [18] and SI [19]. We modify these methods to use task labels during testing for a fair comparison. For the sake of completeness, we additionally compare our approach with different continual learning setting approaches. We compare our approach with replay/memory-based (with/without task labels) approaches such as iCaRL [16], and others. These methods store old task data (additional information) that are used while training for new tasks in order to reduce catastrophic forgetting for older tasks. Note that our method does not store any data from the previous tasks and, therefore, does not need any dedicated memory to store task exemplars.

**Increase in Parameters**: Figure 4 shows the comparison of the growth of model parameters with tasks in the task incremental setting with ResNet-18 architecture. The results indicate that for Progressive Nets [5], the number of parameters increases quadratically with the tasks and has 932.84M parameters after 10 tasks [7]. The rate of growth of parameters is lower for RPSNet [7], but the model still has around 72.26M parameters after 10 tasks. iCARL [16] has around 21.3M parameters after 10 tasks. Our method shows a very insignificant growth of parameters, and the total number of parameters after 10 tasks is also the lowest at 11.8M.

Table 2 reports experimental results for the CIFAR-100 10 classes per task setting on the LeNet architecture. We use the same LeNet architecture (20-50-800-500) as used in [8]. We use task labels during testing for all the methods in Table 2 for a fair comparison. The results indicate that our model CCLL<1,1> performs better than all existing state-of-the-art methods by a significant margin. The APD [8] network has 35% more parameters than the base LeNet network compared to only 0.7% more in our case. We also provide the result for CCLL_BN<1,1>, which is CCLL<1,1> with batch normalization layers added to the LeNet architecture.

**Split CIFAR-10/100**: Figure 5 shows the results for split CIFAR-10/100 task settings using ResNet-32. We compare our model CCLL<1,1> with HNET [6], which is the state-of-the-art for this dataset.

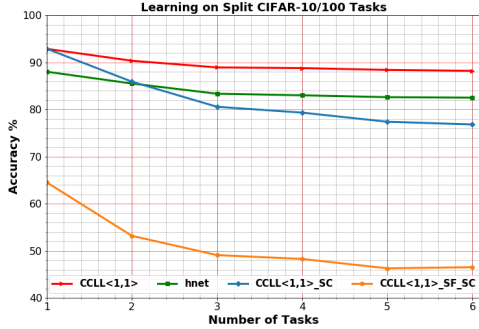

Figure 5: Experimental results on split CIFAR-10/100 using ResNet-32. The reported accuracy for each task is the average of all accuracy values up to that task. SF is a setting where the base module has not been trained at all, and we only train the calibration parameters for each task. SC is a setting where the calibration modules are trained from scratch for each task.

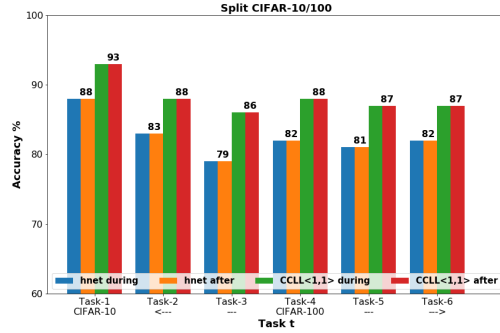

Figure 6: Experimental results on split CIFAR-10/100 using ResNet-32 to check for catastrophic forgetting. The values reported are the achieved accuracy for each task when the network is trained on that task (marked as during) and after the network has been trained on all the tasks (marked as after).

HNET uses task labels during testing, as does our method. We use the same ResNet-32 architecture as used in [6]. The results indicate that our method CCLL<1,1> performs significantly better than HNET and achieves an absolute improvement of 5.7% in the final accuracy. We also provide results for ablation experiments in this setting. We perform ablation for two types of settings - SF (Scratch Freeze) and SC (Scratch Calibration). From the results, we see that the CCLL_SF_SC model performs very badly on the lifelong learning experiment. This is because the calibration parameters are very few and are not enough to learn the tasks properly on their own. This is by design since we do not want to increase the parameters of the network by using heavy calibration modules and instead rely on knowledge transfer from a trained base module. This means that knowledge transfer does happen from our base module that has been trained on the first task and is vital to our model's performance. The results also indicate that the CCLL_SC model performs badly. This means that forward knowledge transfer also happens when we use the calibration weights of the previous task as the initial values of the calibration parameters for the next task and is also vital to our model's performance. This is why we also train the calibration modules for the first task. Figure 6 shows that both CCLL<1,1> and HNET [6] eliminate catastrophic forgetting. But our method achieves higher performance than HNET on the same ResNet-32 architecture. This shows that HNET suffers from performance degeneration, as can be seen from the task-1 (CIFAR-10) accuracy[1] achieved using ResNet-32.

Table 3 reports the experimental results for split CIFAR-10/100 task settings using ResNet-18. We perform ablation experiments to validate the components of our method. The results indicate that removing the spatial calibration module from CCLL<1,1> reduces the final accuracy by around 1.7% absolute value. Removing the channel calibration module also hurts the performance of our model. Removing both SCM and CCM from CCLL<1,1> reduces

Table 3: Experimental results on split CIFAR-10/100 incremental setup using the ResNet-18 architecture. We report the average accuracy of the 6 tasks ($A_6$), % increase in network parameters per task and % increase in total computation. ✓ and ✗ refer to presence and absence respectively.

| $\alpha$ | SCM | CCM | % Params. ↑ per task | % FLOPS ↑ | Accuracy ($A_6$) |
|---|---|---|---|---|---|
| 0 | ✗ | ✗ | 0.0% | 0.0% | 67.06% |
| 0 | ✗ | ✓ | 0.1286% | 0.0009% | 87.85% |
| 1 | ✓ | ✗ | 0.3857% | 0.9955% | 89.06% |
| 1 | ✓ | ✓ | 0.5143% | 0.9964% | 89.58% |
| 2 | ✓ | ✓ | 0.9000% | 1.9919% | 89.94% |
| 4 | ✓ | ✓ | 1.6715% | 3.9829% | 90.45% |
| 8 | ✓ | ✓ | 3.2144% | 7.9650% | 90.99% |

the final accuracy by over 22% absolute value. Therefore, both SCM and CCM are important for our method. The results indicate that using $\alpha$ values greater than 1 leads to better performance, with $\alpha$ equal to 8, showing an absolute increase of about 1.4% over $\alpha$ equal to 1. But we also note that higher $\alpha$ values lead to a higher number of parameters and computations.

Table 4: Large-scale lifelong learning experiments on ImageNet and MS-Celeb datasets. Both have 10 tasks, and the reported accuracy for each task is the average of all accuracies up to that task. '∗' denote memory based approaches.

| Datasets | Methods | 1 | 2 | 3 | 4 | 5 | 6 | 7 | 8 | 9 | Final |
|---|---|---|---|---|---|---|---|---|---|---|---|
| ImageNet-100/10 | LwF [15] | 99.3 | 95.2 | 85.9 | 73.9 | 63.7 | 54.8 | 50.1 | 44.5 | 40.7 | 36.7 |
| | iCaRL∗ [16] | 99.3 | 97.2 | 93.5 | 91.0 | 87.5 | 82.1 | 77.1 | 72.8 | 67.1 | 63.5 |
| | RPSnet∗ [7] | 100.0 | 97.4 | 94.3 | 92.7 | 89.4 | 86.6 | 83.9 | 82.4 | 79.4 | 74.1 |
| | CCLL<1,1> | 99.8 | 99.0 | 99.2 | 98.6 | 98.4 | 98.5 | 98.2 | 97.7 | 97.8 | **97.9**$_{+23.8}$ |
| MS-Celeb-10K/10 | iCaRL∗ [16] | 94.2 | 93.7 | 90.8 | 86.5 | 80.8 | 77.2 | 74.9 | 71.1 | 68.5 | 65.5 |
| | RPSnet∗ [7] | 92.8 | 92.0 | 92.3 | 90.8 | 86.3 | 83.6 | 80.0 | 76.4 | 71.8 | 65.0 |
| | BiC∗ [32] | 95.9 | 96.7 | 96.7 | 96.2 | 95.4 | 94.5 | 93.4 | 91.9 | 90.2 | 88.0 |
| | CCLL<1,1> | 98.3 | 97.9 | 97.7 | 97.7 | 97.8 | 97.8 | 97.7 | 97.7 | 97.7 | **97.7**$_{+9.7}$ |

## 4.4 Experiments on Large-Scale Datasets

On large-scale datasets such as ImageNet and MS-Celeb, the lifelong learning problem is more challenging compared to small-scale datasets. Table 4 shows the results for both ImageNet-100/10 and MS-Celeb-10K/10 settings. We modify LwF to use task labels during testing (same as our method), while the other compared methods are replay/memory-based. The results indicate that our method CCLL performs better than other state-of-the-art methods. Our method CCLL<1,1> outperforms the closest competitor by 23.8% and 9.7% for ImageNet-100/10 and MS-Celeb-10K/10 settings respectively (top-5 accuracy). This margin is significant, especially when considering the fact that these are large datasets. CCLL<1,1> on ImageNet dataset introduces only 0.51% more parameters per task and 0.98% more FLOPS in the model. In case of the MS-Celeb-10K/10 dataset, CCLL<1,1> introduces only 0.35% more parameters per task and 0.98% more FLOPS in the model. Our results are very close to the upper-bound (learning on all the tasks jointly with task labels setting). The upper-bounds on ImageNet-100/10 and MS-Celeb-10K/10 experiments are 98.8% and 98.4% (final accuracy), respectively, which are very close to our reported results (Table 4).

## 4.5 Dependency of CCLL on the first task

To investigate the dependence of our model on the choice of the first task, we run experiments for 5 randomly chosen task orders (with randomly chosen classes per task), each with a different first task, and report the average accuracies. As shown in Table 5, on the datasets we experimented with, our model seems fairly robust against the choice

Table 5: Experimental results on SVHN dataset with ResNet-18. We report the average accuracy of 5 tasks ($A_5$). Calibration modules are trained for each task.

| Base module setting | Base module finetuned on | SVHN ($A_5$) |
|---|---|---|
| Trained from scratch | Only on the first SVHN task | 98.2% |
| Pre-trained on CIFAR-10 | No task (Frozen for all) | 97.9% |
| Pre-trained on CIFAR-10 | Only on the first SVHN task | 98.4% |

of the first task. However, in general, the first chosen task must be large enough and diverse enough to capture a robust and generalizable set of image characteristics (e.g., model filters). If the first task has very few samples per class and/or very few classes, then the performance of CCLL may suffer. Our method is based on re-calibrating the base module to learn new tasks instead of training the full network on new tasks. Re-calibration needs a trained model. Therefore, a randomly initialized (untrained) base module (SF setting as shown in Fig. 5) cannot be expected to work well for any task by using re-calibration. The base module does not have to be trained on the first task and can instead be pre-trained on another dataset. Table 5 shows that a base module pre-trained on CIFAR-10 can be used to successfully perform incremental learning (97.9%) on SVHN using our method by only training the calibration parameters for each SVHN task (CIFAR-10 is significantly different from SVHN). In fact, if we also train the CIFAR-10 pre-trained base module on a randomly chosen first SVHN task, then the incremental performance is even higher (98.4%). However, we do not use a pre-trained base-module in any experiment.

## 5 Conclusion

We propose an efficient lifelong learning method for convolutional neural networks. Through our experiments, we show that our approach outperforms all existing state-of-the-art methods and also introduces a considerably fewer number of additional parameters per task. The model trained with our approach shows no catastrophic forgetting. We also show that forward knowledge transfer plays a vital role in the performance of our approach.

## Broader Impact

Our proposed lifelong learning method is very light-weight and shows no catastrophic forgetting. It will help in improving the performance of models on existing lifelong learning based classification problems. It can also be extended to other applications like image/video segmentation, object detection. Since our method involves calibrating the outputs of the convolutional layers in the model, researchers can use it to convert standard deep learning models to work in lifelong learning settings. For example, a model trained to identify specific crop diseases can be easily extended to also identify rare/new crop diseases restricted to a few regions. Since our approach shows an insignificant increase in parameters, the models produced by our approach will also be more scalable than other methods and thereby better for the deployment of lifelong learning models in light-weight end-user systems. Therefore, both government and non-government entities can provide tailor-made AI services to specific regions/communities in addition to the standard services. This method can be misutilized to perform non-licensed extension to commercially available models. However, we can prevent this by keeping the model architecture encrypted/hidden.

## Acknowledgement

PR acknowledges support from the Visvesvaraya Young Faculty Fellowship. The portion of this research performed at Duke University was supported under the DARPA L2M program.

## Footnotes

[1] https://keras.io/zh/examples/cifar10_resnet/

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
