[Supplementary Material]

# Supplementary Material: Calibrating CNNs for Lifelong Learning

**Pravendra Singh**[*1], **Vinay Kumar Verma**[*2], **Pratik Mazumder**[1],
**Lawrence Carin**[2], **Piyush Rai**[1]
[1]CSE Department, IIT Kanpur, India    [2]Duke University, USA
psingh@cse.iitk.ac.in, vinaykumar.verma@duke.edu,
pratikm@cse.iitk.ac.in, lcarin@duke.edu, piyush@cse.iitk.ac.in

## 1   Additional Results on SVHN

Table 1 reports the experimental results on the SVHN dataset for ResNet-18 and ResNet-18(1/3) architectures. ResNet-18(1/3) is simply ResNet-18 [1], with the number of filters in each layer reduced by 3 times [2]. We use SGD optimizer in all our experiments. In all cases, we run experiments for 5 random task orders and report the average accuracy. From the results, we can see that even with ResNet-18(1/3), which has lesser parameters than ResNet-18, results are comparable for CCLL<1,1> model. CCLL<4,1> with ResNet-18(1/3) performs even better as compared to CCLL<1,1> with ResNet-18.

| Methods | Architecture | 1 | 2 | 3 | 4 | Final ($A_5$) |
|---|---|---|---|---|---|---|
| CCLL<1,1> | ResNet-18 | 98.77 | 98.54 | 98.44 | 98.48 | 98.20 |
| CCLL<1,1> | ResNet-18(1/3) | 98.57 | 98.25 | 98.34 | 98.13 | 98.15 |
| CCLL<4,1> | ResNet-18(1/3) | 98.77 | 98.86 | 98.64 | 98.61 | 98.50 |

Table 1: Experimental results on SVHN dataset with ResNet-18 and ResNet-18(1/3) architectures. There are 5 tasks, and the reported accuracy for each task is the average of all accuracies up to that task.

## 2   Additional Results on CIFAR-100

Fig. 1 shows the experimental results for CIFAR-100 incremental learning tasks using 10, 20 and 50 classes at a time using ResNet-18(1/3) architecture. CCLL with larger values of $\alpha$ such as 2,4,8, performs better as shown in Fig. 1.

## 3   Additional Results on ImageNet-100/10

The results in Table 2 indicate that our method CCLL<4,1> performs better than CCLL<1,1> for ImageNet-100/10. However, CCLL<1,1> introduces 0.51% more parameters per task and CCLL<4,1> introduces 1.66% more parameters per task.

---

[*]Equal contribution.

Figure 1: Experimental results on CIFAR-100 dataset with tasks containing 10, 20 and 50 classes with ResNet-18(1/3) architecture.

| Methods | 1 | 2 | 3 | 4 | 5 | 6 | 7 | 8 | 9 | Final ($A_{10}$) |
|---------|------|------|------|------|------|------|------|------|------|------|
| CCLL<1,1> | 99.8 | 99.0 | 99.2 | 98.6 | 98.4 | 98.5 | 98.2 | 97.7 | 97.8 | 97.9 |
| CCLL<4,1> | 99.2 | 99.2 | 98.9 | 98.9 | 99.0 | 98.9 | 98.6 | 98.5 | 98.6 | 98.7 |

Table 2: Large-scale lifelong learning experiments on ImageNet dataset using ResNet-18 architecture. There are 10 tasks, and the reported accuracy for each task is the average of all accuracies up to that task.