[Reviews · NeurIPS 2020]

Review 1

Summary and Contributions: Update: My initial review noted two main issues with the paper: reliance on the initial model, and the use of task labels during the test phase. The author response addresses the first question, but misses the point on the second one. And this alone is not sufficient to strongly influence my overall rating. In my understanding, several previous methods, such as LwF, iCaRL highlighted in the author response, classify samples without the knowledge of which group of classes (i.e., old or new) they belong to. In other words, they only use a single framework that can identify samples from any of the old or the new classes, without additional information. This is an important distinction when compared with previous work, as it makes the problem somewhat easier --- if we know which group of classes a test sample belongs to, we can use the corresponding version of the learnt model, which in this case is the calibration module. If I'm reading it correctly, the paper and the author response lacks clarity on the use of task labels during the training and the test phase. In my understanding, several (all?) the previous methods considered for comparison, use task labels during the training phase. I'm not sure if any of them use task labels during the test phase (iCaRL and LwF do not do this if I remember correctly). I also do not find support for the response "we also compared our approach with replay/memory-based (with and without task labels) approaches such as iCaRL [16], and others" in the rebuttal nor in the paper. There is only version of iCaRL mentioned in the paper and the rebuttal does not provide additional with/without task labels results. Overall, it appears that the approach needs task labels in the test phase, which is fine, but a strong requirement that needs much more clarity in the paper, along with appropriate comparisons. A clear distinction needs to be made between methods using (or not using) task labels during the test phase. For this reason, I'm not in support of accepting the paper. This paper presents an approach for continual/lifelong/incremental learning in the context of CNN models. Here, the CNN adapts to new tasks/data seen in a sequential fashion, and is capable of handling data from any of the previously seen tasks during the test phase. This is achieved through calibration modules, one for each new task, introduced between each pair of consecutive layers in the original base model. The base model is learnt on the first task (which can be composed of multiple classes), and training data from the other tasks is used only to learn the respective calibration modules. The paper shows that the growth in the number of parameters due to the introduction of these calibration modules in minimal. This approach is evaluated on several datasets.

Strengths: * The paper addresses an important problem. * The proposed approach of using calibration units to handle new data/tasks is quite interesting. It also comes with the advantage of gracefully increasing the size of the updated model. * The method is evaluated on several datasets.

Weaknesses: While the idea of introducing calibration modules is interesting, the paper leaves a few important questions unanswered. * The entire framework relies on having a strong initial model. This depends on the choice of the first task and the classes that it contains. There is no discussion on this important point in the paper. One of ablative studies shows how an untrained initial model completely fails (_SF). This is a good starting point, but insufficient to address concerns on how the model can recover from it. For instance, a scenario where the first task is composed of classes with few examples leads to a weak base model. This leaves very few learnt parameters as part of the calibration modules to represent data from the other (new) tasks. * A second critical point in the paper is the knowledge of task information during the test phase. As noted in lines 155-156, "During testing, depending on the task, the corresponding task-adaptive calibration paramers are used...". Does this imply that the model requires this information on which group of classes a given test sample comes from? This important information is not detailed in the paper. If this additional information is not used by the model, how are the calibration modules learned for each task individually combined during the test phase? Using the task information also makes the method uncomparable to other approaches in the empirical section. In my understanding, EWC, iCaRL, MAS, and several others seen in the comparison do not use this strong information. * An important comparison is missing in the paper. The upperbound of learning on all the tasks jointly helps put the performance of this approach in perspective.

Correctness: Not quite. Several questions on the method and empirical evaluation need to be addressed (see above).

Clarity: Needs further clarification (see above).

Relation to Prior Work: A few key differences need to highlighted more clearly.

Reproducibility: No

Additional Feedback: * For a more complete picture, it is necessary to compare with all the methods on the all the datasets. For example, including iCaRL in comparisons shown in Tables 1 and 2 helps get a full story. * An additional ablative study with CCLL_SF only in Figure 5 could also be useful. * Showing the evolution of performance on each individual class (and also task) over time will also be interesting to better understand the method.


Review 2

Summary and Contributions: This paper proposes a calibration approach for continual/lifelong learning. It shows that the activation maps generated by the CNN trained on the old task can be calibrated using very few calibration parameters for new tasks. The proposed approach stores the task-adaptive calibration parameters to avoid catastrophic forgetting. The proposed method achieves a new state of the art on four benchmark datasets.

Strengths: - The paper is well written and easy to follow. - The idea is straightforward and relatively new. The model description is very clear. - The experimental evaluation is thorough. Rich experiments and evaluation.

Weaknesses: 1. Some important references are missing. "Piggyback: Adapting a Single Network to Multiple Tasks by Learning to Mask Weights", ECCV 2018. This paper studies a pruning based approach where multiple tasks share the same pre-trained network but with a different pruning pattern. Only the pruning pattern is stored to avoid catastrophic forgetting. Conceptually, this approach is quite similar to the proposed method. 2. The paper claims in line 78-79 that "the calibration parameters of the previous task also serve as good initial weights for learning the calibration parameters of the new task". This claim is quite strong but not further justified in the later sections.

Correctness: Yes and Yes.

Clarity: Yes.

Relation to Prior Work: Yes.

Reproducibility: Yes

Additional Feedback: I went through the rebuttal and I will keep my original scores.


Review 3

Summary and Contributions: (Post-rebuttal) The rebuttal addressed my concerns. I am still in favor of this paper being accepted. This paper presents an approach for lifelong/continual learning of convolutional neural networks (CNNs) that does not suffer from the problem of catastrophic forgetting when moving from one task to the other. The paper shows that the activation maps generated by the CNN trained on the old task can be calibrated using very few calibration parameters, to become relevant to the new task. Based on this, the approach calibrates the activation maps produced by each network layer using spatial and channel-wise calibration modules and trains only these calibration parameters for each new task in order to perform lifelong learning. The calibration modules introduce significantly less computation and parameters when compared to approaches that dynamically expand the network. Over an extensive series of experiments, the proposed approach is shown to be superior to a large variety of comparison approaches.

Strengths: Positives: + This is a strong neural network architecture paper addressing the important open problem of lifelong learning. The proposed approach is novel, and this paper makes a good contribution to this research area. + This paper contains an extensive empirical evaluation that considers the SVHN, CIFAR, ImageNet, and MS-Celeb datasets and a good number of state-of-the-art comparison approaches. The proposed approach shows decisive improvement over existing methods – in some cases with impressive measures of improvement. + The paper is very well written and easy to follow. This work could be reproduced from what is written.

Weaknesses: Negatives: - One potential problem: Lines 147-148 state “Through an ablation experiment, we show that if we do not train the base module at all and only train the calibration parameters for each task, the network performance is hurt drastically.” This seems to imply that the training for the first task needs to be good, otherwise you may end up with a problem similar to catastrophic forgetting down the line. How can one determine that the training of the base module was effective?

Correctness: The claims and method are correct. The empirical methodology is exceptionally strong.

Clarity: The paper is very well written and easy to follow. This work could be reproduced from what is written.

Relation to Prior Work: Related prior work is described in detail.

Reproducibility: Yes

Additional Feedback: Other Comments: Move Sec. 3 (the related work section) to Sec. 2. Having related work in between the method and experiments breaks up the flow for the reader. No error is reported for any of the experiments. It would be nice to see how stable these results are (i.e., how significant a change in the random seed is or what happens if you shuffle the data). The broader impact statement is very terse and doesn’t add anything to the paper. Will the code be released after publication?


Review 4

Summary and Contributions: This paper proposes a new method for continual learning of convolution networks that doesn't suffer from catastrophic forgetting. In particular, a) It proposes calibration of activation maps after each layer using specific spatial and channel-wise calibration modules that learn task-specific knowledge. b) It requires a very small number of new parameters per task. c) It can perform significantly better than existing state-of-art methods and successively mitigate catastrophic forgetting problems.

Strengths: + Nice well-written paper. + The proposed calibration modules are simple and can be easily added to convnet layers. + Show effective forward knowledge transfer on new tasks. + Sufficiently detailed experiments and comparison to prior methods. + The overall idea seems quite simple but is able to perform significantly better than existing state-of-art. This is a good plus for the continual learning area using occam's razor.

Weaknesses: - Experiments are limited to classification tasks. Recently, datasets like CORe50 with more complicated tasks like object recognition/detection for continual learning have been released. - There should be a mention that the method is limited to continual learning (CL) with task labels setting, instead of a broader claim for the whole area of CL. Without tasks labels (i.e. any indication of when and how the task changes) the method will not be able to choose the correct calibration module for the new task and fail to work. I acknowledge that this distinction of with and without task labels has been missing in prior work, and won't be a significant factor for my review. However, new works in CL should make clear the distinguishment for further development of this area.

Correctness: Yes, the claims and methods are correct and emperically sound.

Clarity: The paper is well written and has clarity in presenting and motivating ideas. I did not find any noticeable grammatical/formatting issues.

Relation to Prior Work: Yes, there is ample discussion to prior work and how this paper differs from them.

Reproducibility: Yes

Additional Feedback: Please improve on my suggestions in the weakness section. Great work overall!

[Author Response · NeurIPS 2020]

We appreciate the time invested by the reviewers in carefully reading our paper and providing very helpful and detailed
comments. We sincerely thank the reviewers for their positive feedback on: paper writing [R3, R4, R5], extensive
empirical evaluation [R1, R3, R4, R5], novelty of our idea [R1, R3, R4, R5], and its simplicity [R5].

**Dependence of our approach on the first task [R1, R4]:** Our
method does not require choosing "good" classes for the first
task or choosing a "good" first task. As mentioned in the paper
(Line# 229), we run experiments for 5 randomly chosen task
orders (with randomly chosen classes per task) and report the
average accuracy. Therefore, our results are an average of
different task orders with different first tasks.

Table 1: Experimental results on SVHN dataset with ResNet-18. We report the average accuracy of 5 tasks ($A_5$). Calibration modules are trained for each task.

| Base module setting | Base module finetuned on | SVHN ($A_5$) |
| --- | --- | --- |
| Trained from scratch | Only on the first SVHN task | 98.2% |
| Pre-trained on CIFAR-10 | No task (Frozen for all) | 97.9% |
| Pre-trained on CIFAR-10 | Only on the first SVHN task | 98.4% |

Our method is based on re-calibrating the base module to learn new tasks instead of training the full network on new
tasks. Re-calibration needs a trained model of some sort. Therefore, a randomly initialized (untrained) base module
(_SF) cannot be expected to work well for any task by using re-calibration. The base module does not have to be
trained on the first task and can even be pre-trained on another dataset. We have shown in Table 1 above that a base
module pre-trained on CIFAR-10 can be used to successfully perform incremental learning ($A_5$: 97.9%) on SVHN
using our method by only training the calibration parameters for each SVHN task. In fact, if we also train the CIFAR-10
pre-trained base module on a randomly chosen first SVHN task, then the incremental performance is even higher ($A_5$:
98.4%). Please note that CIFAR-10 is significantly different from SVHN. *Please also note that in our paper, we have
not used a pre-trained base-module in any experiment.*

If the data setting is such that classes have few samples, then our method can be modified to use the pre-trained
base module setting described above. However, this is a different problem setting in itself. We will include all the
above-discussed points in the revised version. Our response to other comments from reviewers is provided below.

——————— **Reviewer 1 (R1)** ———————

**Comparison with other approaches:** Our approach uses task labels and follows the continual learning with task
labels setting. We compared our approach with methods (APD [8] ICLR-20, HNET [6] ICLR-20, GEM [36], LwF
[15], and several others) that use same task labels setting. For the sake of completeness, we *additionally* compared
our approach with various other continual learning setting approaches. In particular, we also compared our approach
with replay/memory-based (with and without task labels) approaches such as iCaRL [16], and others. These methods
store old task data (additional information) that are used during training for new tasks so that catastrophic forgetting is
reduced for older tasks. Please note that our method does not store any data from the previous tasks and, therefore, does
not need any dedicated memory to store task exemplars. We will mention all these details for all the compared methods
in the revised version. The numbered citations refer to the references of the main paper.

**Upper-bound:** Our results are very close to the upper-bound (learning on all the tasks jointly with task labels setting).
The upper-bounds on ImageNet-100/10 and MS-Celeb-10K/10 experiments are 98.8% and 98.4% (final accuracy),
respectively, which are very close to our reported results for these experiments. We will include the upper-bounds for
all the experiments in the revised version.
**Additional suggestions:** We will incorporate all your suggestions in the revised version.

——————— **Reviewer 3 (R3)** ———————

**Piggyback:** We will add more discussion on the Piggyback method (reference [5]) in the revised version.
**Paper claims in line 78-79:** We have justified this claim using an ablation study of setting SC (Scratch Calibration), as
mentioned in Lines# 280-282. We will further highlight this point in the revised version.

——————— **Reviewer 4 (R4)** ———————

**No error bar is reported:** We run experiments for 5 random task orders and report the mean accuracy in the paper for
all experiments. We will also mention variance in the revised version. Thanks for your suggestion.
**Broader impact statement:** We will improve it in the revised version. The code will be released after the publication.

——————— **Reviewer 5 (R5)** ———————

**Experiments on object recognition/detection:** We will include results on these tasks in the revised version.
**Method setting:** Thanks for pointing this out. Yes, our approach is limited to continual learning (CL) with task labels
setting. We will be happy to mention it in the revised version.

[Meta-Review · NeurIPS 2020]

The paper proposes a continual learning approach for CNN models. This is achieved through spatial and channel-wise calibration modules, one for each new task. These calibration modules are introduced between each pair of consecutive layers in the original base model. The base model is learnt on the first task, and training data from the subsequent tasks is used to learn the calibration modules. Extensive experiments show the superiority of the proposed method in terms of accuracies, with minimal computation and storage overhead. It is important to emphasize that the proposed approach requires task labels in the test phase. This is a strong requirement that needs much more clarity in the paper, along with appropriate comparisons. If we know which group of classes a test sample belongs to, we can use the corresponding version of the learnt model, which in this paper is the calibration module. A clear distinction needs to be made between continual learning methods using or not using task labels during the test phase. We strongly suggest the authors to incorporate these suggestions into their final version of the paper.